

# The effects of occupational disruption during COVID-19 lockdowns on health: a cross-sectional study

Bernard Austin Kigunda Muriithi[1] and  R. Curtis Bay[2]

[1] Communication Disorders and Occupational Therapy, University of Arkansas Fayetteville & University of Arkansas for Medical Sciences, Fayetteville, AR, United States of America
[2] Interdisciplinary Health Sciences, A. T. Still University of Health Sciences, Mesa, AZ, United States of America

## ABSTRACT

The disruption in daily activity performance during COVID-19 lockdowns is widely understood to have impacted health, but a better understanding of how restricted performance of specific activities are associated with health is needed. This cross-sectional study answers the following question: How were changes in the performance of 16 daily activities associated with health during COVID-19 lockdowns? A total of 116 participants completed an online survey rating their health before and during COVID-19 lockdowns and comparing their recollection of the performance of 16 activities before COVID-19 with their performance during lockdowns. Multiple stepwise linear regression analysis was used to estimate the relationship between self-reported changes in activities during lockdowns and concurrent (during-lockdown) health status, while controlling for pre-COVID-19 health status. Only changes in activities that were uniquely and significantly associated with lockdown health status were retained in the final model. Health before COVID-19 accounted for 3.7% ($P = 0.039$) of the variance in health during COVID-19 lockdowns. After controlling for health before COVID-19, five types of activity were significantly and uniquely predictive of health during lockdowns, together accounting for 48.3% of the variance. These activities and the variances they accounted for were rest and sleep (29.5%, $P < 0.001$), play and recreational activities (8%, $P < 0.001$), work (4.8%, $P = 0.002$), personal hygiene (3.2%, $P = 0.01$), and healthy eating (2.8%, $P = 0.013$). The study suggests that these five types of activity should be prioritized in policy or interventions when participation in activity is constrained by lockdowns or comparable factors.

# INTRODUCTION

The declaration by the World Health Organization (WHO) of COVID-19 as a global pandemic, on 11 March 2020, resulted in an unprecedented global decline in participation in daily activities. Countries around the world responded to the pandemic by requiring social distancing, restricting movement, and lockdowns to curb the spread of the disease. Lockdown measures included closing schools, public facilities, shops, restaurants, bars,

Corresponding author
Bernard Austin Kigunda Muriithi,
muriithi@uark.edu

parks, and airports. There was an abrupt and sharp decline in participation in essential daily activities such as working, attending schools, leisure and recreational activities, participation in communal worship, and socializing, among others. The inability to participate in normal daily activities had far-reaching health implications, but more studies are needed to shed light on how reduced performance in specific activities affected specific health domains.

Occupational therapists, who view activity performance as the means and ends of intervention (*McLaughlin Gray, 1998*), consider all types of meaningful things people do as *occupations*. Occupations occupy time with meaningful things people need to, want to, or are required to do (*World Federation of Occupational Therapists, 2024*). The American Occupational Therapy Association (AOTA) publication "Occupational Therapy Practice Framework, version 4" (OTPF-4) defines occupations as "everyday activities that people do as individuals, in families, and with communities to occupy time and bring meaning and purpose to life" (*American Occupational Therapy Association, 2020*). Occupations are classified into activities of daily living (ADL), instrumental activities of daily living (IADL), health management, rest and sleep, education, work, play, leisure, and social participation. However, the classification has been argued to limit the scope and thinking of occupational therapy practitioners and researchers (*Hinojosa, 2007*). For instance, OTPF-4 does not include numerous non-sanctioned activities (*e.g.*, drug addiction and violent behaviors), resulting in the false portrayal of occupations as only positive, productive, and healthy activities. This assumption has been challenged by the growing recognition that occupations often have a dark side (*Twinley, 2013*; *Twinley, 2021*), particularly when occupations that include addictive activities, impulse control disorders, and violence behaviors are also considered meaningful occupations (*Kiepek & Magalhães, 2011*; *Twinley & Addidle, 2012*).

A temporary disruption in participation in everyday activities is termed "occupational disruption" (*Whiteford, 2000*). This concept has been used to characterize the changes in occupational participation that resulted from COVID-19 restrictions throughout the world (*Asbjornslett et al., 2023*; *Engels, Segaux & Canouï-Poitrine, 2022*; *Jokic & Jokic-Begic, 2022*; *Keptner, Ekelman & Milliken, 2023*; *Rotenberg et al., 2021*; *Segev-Jacubovski & Fogel, 2022*; *Wegner et al., 2022*). Occupational disruption refers to "a state of impaired performance, which usually resolves itself given supportive conditions" (*Whiteford, 2000*, p. 201). There is extensive evidence that restricted participation in activities during the COVID-19 pandemic affected health, but it is not clear how occupational disruption, with respect to a specific type of occupation, affected specific health domains such as quality of life, resilience, and mental health.

## How occupational disruptions influenced health

The reviewed literature shows that work, a specific type of everyday activity for many people, was severely affected by the COVID-19 lockdowns. "Work" is often understood in English as synonymous with occupation, but in occupational therapy, work is just one type of human occupation, an activity distinguished by its ability to generate income, but not always (*American Occupational Therapy Association, 2020*). In common usage, the term work is associated with terms such as employment, career, work place, and remuneration; but even within this framework, it is a complex concept that incorporates innate, relational,

well-being, identity, and learning dimensions (*Laan, Ormsby & McIlveen, 2023*). Due to the pandemic, there were severe job losses (*Mamgain, 2021*; *Munawar et al., 2021*; *Posel, Oyenubi & Kollamparambil, 2021*; *United States Bureau of Labor Statistics, 2021*; *Zhang et al., 2020*) that affected the affordability of necessities such as food and clothing. Job losses may have led to loss of identity, loss of meaning or purpose in life, and a lack of something meaningful to do with time. It is not surprising that these conditions led to serious mental health problems, disproportionately affecting underprivileged groups such as racial minorities and people with lower education or lower socioeconomic status (*Matthews et al., 2021*).

Another type of activity that was significantly affected by the lockdowns was education, which involves activities that occupy most of the day for many children and youth. At its peak, the pandemic resulted in school closures in 188 countries, affecting more than 1.6 billion children and youth (*UNICEF, 2022*). Although many schools soon implemented distance learning platforms for academic continuity, quality and access were challenged by existing inequalities (*Karvounides et al., 2021*; *Walker & Koralesky, 2020*) and limited time to prepare faculty and students with appropriate technologies (*Salas-Pilco, Yang & Zhang, 2022*). Furthermore, some studies have shown that students prefer in-person classroom activities because they are more supportive of their psychological and physical well-being (*Shrivastava et al., 2022*; *Spencer & Temple, 2021*). Online learning presented predictable but unavoidable challenges, including behavioral engagement due to perceived or real limitations in opportunities to interact with peers (*Salas-Pilco, Yang & Zhang, 2022*), cognitive engagement as online learning required enhanced methods to keep learners motivated (*Binali, Tsai & Chang, 2021*), and affective engagement due to high negative perceptions of the online learning experience (*Ferrer et al., 2022*). Online learning also caused a disruption in normal daily activity routines. Collectively, these factors appear to have produced the noticeable increase in emergency room hospital visits due to mental health conditions among children (*Lee, 2020*; *Leeb et al., 2020*).

In all age groups, leisure and recreational activities, eating out, shopping and socializing were significantly affected due to the closure of public recreational facilities, restaurants and bars, stores, movie theaters, community centers, and parks. Travel was limited due to cancellations of flights and state or national restrictions, making it impossible for people to see their loved ones for extended periods. In particular, lockdowns deprived people of participation in activities that bring about some level of resolution after losing loved ones, at a time when many lost family and friends due to the pandemic, causing a range of negative emotions and distress to linger indefinitely (*Boss, 2022*), an experience that has been linked to physical and psychological health problems (*Selman et al., 2020*). The visitation restrictions during COVID-19 made it impossible to participate in usual hospital care during episodes of critical illness and the opportunity to participate in normal grief activities associated with death and bereavement (*Jordan et al., 2022*). This created a sense of helplessness in seeing loved ones die alone (*Hernández-Fernández & Meneses-Falcón, 2021*), which likely produced self-blame, a powerful predictor of grief difficulties (*Stroebe et al., 2014*).

Activities such as work, education, recreation, and hospital visits have some components of physical activity. However, studies have also shown that physical activity, as a specific activity type, was reduced during the COVID-19 pandemic by more than 25% (*Bu et al., 2021*; *Meiring et al., 2021*; *Meyer et al., 2020*; *Souza et al., 2023*; *Tison et al., 2020*). The decline in physical activity during the pandemic has been associated with a decline in mental health and well-being (*Chi et al., 2021*; *Faulkner et al., 2022*).

The unprecedented, global, and prolonged duration of disruption of normal daily activity, along with the documented effects on health, reaffirm the belief that occupation is a determinant of health (*American Occupational Therapy Association, 2017*). Some studies have shown how occupational disruptions affected specified domains of health. For example, occupational disruption has been reported to produce serious negative effects on physical health (*Kiuchi, Kishi & Araki, 2020*; *Marconcin et al., 2022*), mental health (*Chien, 2022*; *De Oliveira et al., 2023*; *Marconcin et al., 2022*), psychological resilience (*Karimi Khordeh et al., 2022*; *Killgore et al., 2020*), social relationships (*Long et al., 2022*; *Ottoni, Winters & Sims-Gould, 2022*; *Philpot et al., 2021*), and quality of life (*De Oliveira Almeida et al., 2023*; *De Oliveira et al., 2023*; *Kan et al., 2023*; *Kandula & Wake, 2021*). However, such studies have been limited to one or a few domains.

The emerging concept of occupational resilience was proposed to encapsulate the capacity of individuals to persist in occupations when faced with adversity (*Muriithi & Muriithi, 2020*). Occupational resilience is "the capacity for persistence in the performance of an occupation, whereby capacity is reflected in both duration and intensity of engagement in a specified occupation" (*Muriithi et al., 2022*). This ability to persist, with respect to COVID-19 lockdown restrictions, denotes an ability to respond in ways that enable participation in occupations despite restrictions. Occupational resilience was used to characterize positive responses that produced continued participation in occupations despite the restrictions of COVID-19 (*Brown, 2021*), but individuals, groups, communities, and countries had different capacities to respond. It has been argued that occupational resilience is measurable (*Muriithi & Gore, 2023*) and modifiable through clinical interventions (*Muriithi et al., 2022*).

## Study objective

The current study aimed to answer the following question: *How did changes in the performance of 16 everyday activities influence health [overall health, resilience, mental health, physical health, quality of life, and social well-being] during the COVID-19 lockdowns?* Occupational disruptions resulting from the COVID-19 pandemic provided a unique opportunity to re-examine the effects of everyday activity performance on health. It was expected that the relationship between different types of activity and health would differ in nature (*i.e.,* either positive or negative correlation) and strength (*i.e.,* performance of some activities would correlate more strongly with health than others). An understanding of how different activities influence health can inform clinicians, researchers, and public health officials on possible areas of focus, so that there can be more targeted interventions to address health in comparable circumstances in the future.
## METHODS

### Ethical approval

This study was approved by the Institutional Review Board of A.T. Still University of Health Sciences as Exempt Protocol #2021-202 (minimal risk).

### Design

The study used a cross-sectional design. Sixteen activity types, selected and adapted from OTPF-4 (*American Occupational Therapy Association, 2020*), were included in the survey. The first section of the survey (Table 1) requested basic demographic information (including age, gender, race/ethnicity, country of residence, and employment status) but without personally identifiable information. The second section of the survey (Table 1) asked participants to rate six health domains (*i.e.,* overall health, resilience, quality of life, physical health, mental health, and social well-being) for two time periods: their status before the pandemic and their status during the COVID-19 lockdowns. The third section (Table 1) asked participants how their performance in each of the 16 types of activity during lockdowns compared to their performance before the pandemic. In this section, the choices were as follows: *Much Worse than Before, Worse than Before, Same as Before, Better than Before*, and *Much Better than Before*. The Qualtrics online platform was used to collect data. The link to the survey required each participant to electronically (1) confirm that they were adults (18 and older), and (2) provide consent electronically by selecting "I agree to participate", before accessing the survey. The survey was tested before distribution using an online link and QR code. The invitation to participate was distributed through University emails and investigators' Facebook, LinkedIn, and Twitter (X) accounts.

### Statistical analysis

Summary statistics are provided, including means (standard deviations) and counts (percentages), as appropriate. We calculated health status before COVID-19 as the mean of responses to questions in five health domains (*i.e.,* overall health, ability to meet responsibilities, quality of life, physical health, and mental health) using a 5-point Likert-type. We formed an analogous scale using these same questions for the COVID-19 lockdown period. Cronbach's alpha and principal components analysis were used to evaluate the internal consistency and dimensionality of both pre- and during COVID-19 scales. Multiple linear stepwise regression analysis was used to estimate the relationship between self-reported changes in activities during lockdowns and during-COVID-19 health status, while controlling for pre-COVID-19 health status.

During-COVID-19 health status served as the criterion variable. Pre-COVID-19 health domain score was entered as the first predictor variable in the regression analysis, and retained. Then, in the second block of the analysis, we entered the 16 variables reflecting the change in activities during lockdowns in a stepwise manner. This analysis yielded estimates of the degree to which a subset of self-reported changes in activities during COVID-19 were associated with health status, while controlling for differences in health status among respondents pre-COVID-19. Only changes in activities that were uniquely and significantly associated with lockdown health status were retained in the final model.

**Table 1  Survey questions.**

| Demographics | | | |
| --- | --- | --- | --- |
| Age | Gender—M (male) F (female) NB (Nonbinary) | Race/Ethnicity | Country of residence |
| Employment Status: | Employed (E) | Unemployed (U) | Student (S) |
| Health Attributes | | | |
| Very poor                Poor                Average | | Good | Very good |
| Using the scale above | | Before COVID-19 | During COVID-19 Lockdowns |

Describe your overall health

Describe your ability to live as usual and meet your responsibilities.

How was your quality of life?

How was your physical health?

How was your mental health?

How were your social relationships?

*For each of the following activities, indicate how well you were able to perform the activity during COVID19 lockdowns compared to before the pandemic.*

| Much worse than before        Worse than before        Same as before | | Better than before | Much better than before |
| --- | --- | --- | --- |

Personal hygiene activities *e.g.*, taking showers or baths, brushing teeth, grooming, and dressing.

Sleep and rest.

Healthy eating.

Activities to satisfy sexual needs.

Activities involving caring for children, other people, or pets.

Activities to maintain house, yard, garden, house plants, appliances, vehicles, and repairs.

Religious or spiritual activities *e.g.*, prayer, meditation, reading the bible, Koran, or other religious books.

Shopping (in-store or online).

Physical activities like exercise, jogging, hiking, running, or walking outside.

Participating in classes, programs, or activities for education or skills development.

Work for pay.

Volunteering to serve others (that is, working without pay)

Play or recreational activities.

Engaging in activities that result in interacting with others in social networks, religious or spiritual groups.

Engaging in group activities with family, friends or peers who support each other.

Engaging in activities to give and receive affection, not necessarily including intimacy or sexual activity.

An alpha of .05 (two-tailed) was used as the criterion for statistical significance. SPSS ver. 28 (IBM Corp., Armonk, NY, USA) was used for statistical analysis.

## RESULTS

### Participants

A total of 131 participants completed the survey. Nine respondents were removed because they were not from the United States, and six were removed because they did not provide complete data, leaving 116 participants in the analysis dataset. The participants were predominantly female (90%). The average age (SD) was 31.3 (10.8) years, ranging from 22 to 68 years. In racial composition, they were 74.1% White, 7.8% Hispanic, 6.9% Asian, 6.0% Black, and 5.2% identifying as 'other'. The overall employment status among the
**Table 2** Activities contributing significantly to the mean health status during COVID-19 lockdowns.

| Factor (or activity) | Percentage of Variance Accounted for ($R^2$) | *P*-value for Change in $R^2$ |
|---|---|---|
| Overall health/quality of life before COVID-19 | 3.7 | 0.039 |
| Sleep and rest | 29.5 | <0.001 |
| Play or recreational activities | 8.0 | <0.001 |
| Regular work for pay | 4.8 | 0.002 |
| Personal hygiene | 3.2 | 0.010 |
| Healthy eating | 2.8 | 0.013 |

participants changed from 63.8% before COVID-19 to 43.1% during the COVID-19 lockdowns. The percentage of participants who were students increased from 33.6% before the pandemic to 45.7% during lockdowns. Unemployment status increased among participants from 2.6% before COVID-19 to 11.2% during COVID-19 lockdowns.

We calculated a single health status value from the mean of responses about health status before COVID-19 in five health domains: overall health, ability to meet responsibilities, quality of life, physical health, and mental health. We formed an analogous scale using these same questions for the COVID-19 lockdown period. These five items for both pre- and during-COVID-19 time periods were highly intercorrelated (Cronbach's alpha for Pre = .88, and Cronbach's alpha for During = .84). Principal components analysis yielded single-dimension solutions for both Pre (all component loadings >.79) and During COVID-19 (all component loadings >.76).

As shown in Table 2, overall health before COVID-19 accounted for 3.7% of the variance in overall health during COVID-19 lockdowns. After controlling for health before COVID-19, five activities were significantly and uniquely predictive of health during COVID-19 lockdowns, accounting for an additional 48.3% of the variance.

## DISCUSSION

The purpose of this study was to determine how the changes in daily activity performance, resulting from COVID-19 lockdowns, affected health. Of the sixteen selected types of daily activities included in the survey, changes in five of them during lockdowns significantly correlated with changes in health. The results support our hypothesis that changes in participation in different types of activities would impact health differently depending on the type of activity. Presumably, higher occupational resilience produced better participation in certain activities during COVID-19 lockdowns for some individuals, groups, or communities, but not for others. Subsequent variations in occupational participation contributed to variations in health outcomes between people. The results showed that some types of activity significantly influenced health more than others; therefore, these activities should have priority in policies or other interventions.

The type of activity that affected health the most was rest and sleep, contributing 29.5% of the variance between pre-COVID-19 and lockdown health status. It can seem an oxymoron

to describe rest and sleep as occupation, because they often involve minimal consciousness, but the two meet the criteria for occupations (*American Occupational Therapy Association, 2020*), being things people do that occupy time and are meaningful. Rest and sleep revitalize brain function, making the performance of other occupations possible or better. Rest involves activities that produce a relaxed state in which demanding physical, mental, or social activities are reduced (*American Occupational Therapy Association, 2020*). Sleep is a state of immobility with reduced responsiveness that enables recovery from the accumulated debt of sleep that builds during wakefulness (*Siegel, 2005*). Restoration of neural functioning enables rest and sleep to promote physical and emotional health (*Khazaie et al., 2023*; *Matricciani et al., 2019*) and improves participation in other activities such as work (*Deng et al., 2022*; *Litwiller et al., 2017*) and education (*Bono & Hill, 2022*). Rest and sleep, by far, more significantly influenced health than any other type of occupation.

The results suggest that changes in play and recreational activities between pre-COVID-19 and lockdowns were associated with significant changes in health status, contributing 8% in variance. This was an expected outcome because the lockdowns forced a reduction in recreational activity options, and these activities are associated with positive subjective well-being (*Cummings, 2002*) and mental health (*Sala et al., 2019*; *Takiguchi et al., 2023*; *Yoshida et al., 2021*; *Zulyniak et al., 2020*). Play and recreational activities are often leisure activities, performed when people are free from essential daily activity obligations like work or education. They are done for relaxation, enjoyment, and acquiring skills, among other reasons. Leisure activities have been effective in reducing anxiety, stress, or pain in people in distress, such as those in hospitals (*Adam-Castelló et al., 2023*). As play and recreational activities differ substantially, their effects on health could vary significantly, as do the mechanisms of action through which they affect health. There are more than 600 such mechanisms, classified under psychological, biological, social, and behavioral processes (*Fancourt et al., 2021*).

Play and recreational activities are often social, as they are commonly performed with or in the company of other people. But, consistent with the AOTA classification of occupations, social participation was a separate category in the survey. Contrary to our expectation, social participation by itself did not significantly impact health during the lockdowns. Social participation is a concept used to describe "activities that involve social interaction with others, including family, friends, peers, and community members, and that support social interdependence" (*American Occupational Therapy Association, 2020*). It is the key type of activity most significantly impacted by social distancing policies. Social participation has known effects on health, and interventions were recommended to help people maintain social contacts during similar pandemics (*Sikali, 2020*), as positive social contacts are associated with psychological and physical health (*Baumeister & Leary, 1995*), and loneliness has been linked to suicidal behavior in the general population (*Stickley & Koyanagi, 2016*). The fact that changes in social participation did not significantly impact lockdown health status was therefore a surprising outcome.

Pre-COVID-19 *versus* lockdown physical activity [for example exercise, jogging, hiking, running, or walking outside] changes did not show a strong association with changes in health status during lockdowns. Physical activity was classified separately from play

and recreation, although many play and recreational activities are physical activities. This was done to help detect any effect of physical activity by itself, but contrary to what other studies found (for example, *Chi et al., 2021*; *Killgore et al., 2020*), there was no significant association between change in physical activity and change in health status during lockdowns.

As the type of activity associated with income generation, work was expected to be among the activities that would significantly impact health during lockdowns. Pre-COVID-19 *versus* lockdown changes in work performance contributed 4.8% of variance between pre-COVID-19 and lockdown health status. It was not a surprising outcome, as unemployment has been attributed to psychological distress in a study involving a much larger sample size (*Matthews et al., 2021*), and psychological resilience, which was significantly impacted by work, influences mental health (*Killgore et al., 2020*) and coping skills (*Vinkers et al., 2020*). Work also provides financial resources for necessities of daily living, including costs associated with play and recreation among other essential daily activities. Therefore, work likely impacted health directly, but it may also have indirectly impacted health by limiting participation in other important activities due to lack of financial resources.

Changes in the performance of personal hygiene activities (*e.g.*, taking showers or baths, brushing teeth, grooming, and dressing), called activities of daily living (ADL), significantly contributed to health during lockdowns, accounting for 3.2% variance between pre-COVID-19 and lockdown health status. The impact of participation in ADLs on health is consistent with outcomes of a systematic review that found that poor participation in ADLs, due to the COVID-19 disease, resulted in negative health-related quality of life (*De Oliveira Almeida et al., 2023*). The interruption of routines produced significant variations in the performance of personal hygiene activities, which are commonly performed in patterns that punctuate other daily activities. Lockdowns interrupted essential routines that make the performance of these activities automatic or regular.

Lastly, healthy eating accounted for 2.8% variance between pre-COVID-19 and lockdown health status. Eating is an essential everyday activity that supplies nutrients to the bodies, but in the survey, this activity was framed positively as 'healthy eating', not just 'eating'. Eating healthy is distinguished in *Health People 2030* as an important activity for meeting health goals set by the *U.S. Department of Health and Human Services (2024)*. But eating as an activity can also be detrimental to health if the food consumed is not the right amount or variety. Unhealthy eating has been a contributor to conditions like obesity, cardiovascular disease, hypertension, stroke, type 2 diabetes, metabolic syndrome and some cancers (*Gropper, 2023*). Healthy eating, on the other hand, can treat or help manage some chronic health conditions (*Di Renzo et al., 2019*; *Slawson, Fitzgerald & Morgan, 2013*). It is therefore not surprising that healthy eating correlated positively and significantly with lockdown health status.

## STUDY LIMITATIONS

The primary limitation of this study was that it relied on participants' recollection of their occupational performance and health status before COVID-19, which may not have been

perfectly accurate in light of their altered circumstances during lockdown. Furthermore, the study relied heavily on the AOTA list of occupations and ignores numerous other activities that people do, which could have significant health effects in positive or negative ways. For example, consideration of so-called 'dark occupations' may have revealed whether participation in such activities increased during COVID-19 lockdowns and whether they were predictive of health. Furthermore, the study relies on a relatively small and homogeneous sample, consisting predominantly of white American females. Further exploration of the relationship between health and participation in various occupations is warranted. Future studies are recommended to include a wider range of occupations, as well as larger and more diverse samples, including different socioeconomic, cultural, age and political factors.

## CONCLUSION

This study revealed that occupational disruptions from COVID-19 restrictions impacted health differently depending on the type of activity considered. The five types of activity which impacted health the most were rest and sleep, play and recreational activities, work, personal hygiene, and healthy eating. Further research is needed to better reveal the effects of specific activities on health, particularly during pandemics or comparable occurrences that disrupt performance of everyday activity.

### Funding
The authors received no funding for this work.

### Competing Interests
The authors declare there are no competing interests.

### Author Contributions
- Bernard Austin Kigunda Muriithi conceived and designed the experiments, performed the experiments, analyzed the data, prepared figures and/or tables, authored or reviewed drafts of the article, and approved the final draft.
- R. Curtis Bay analyzed the data, prepared figures and/or tables, authored or reviewed drafts of the article, and approved the final draft.

### Human Ethics
The following information was supplied relating to ethical approvals (*i.e.*, approving body and any reference numbers):
A.T. Still University of Health Sciences as Exempt Protocol #2021-202 (minimal risk).

### Data Availability
The raw data is available in the Supplemental File.

## Supplemental Information

Supplemental information for this article can be found online at http://dx.doi.org/10.7717/peerj.17594#supplemental-information.

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
