# Peer review of "The effects of occupational disruption during COVID-19 lockdowns on health: a cross-sectional study"

_PeerJ, doi:10.7717/peerj.17594_

## Round 0.1 · original submission · Major Revisions

I do agree with the statistical concerns raised by both R1 and R2. Please address all these issues, as well as the concerns raised by R3 on the timing of the data collection.

Reviewer 1 ·

Basic reporting

no comment

Experimental design

- It is rare to include participants from different counties in the same study, especially from counties have very different covid-19 incidence rates, death rates and policy. Including participants from different counties also makes it hard for other researchers to interpret and generalize your study results. There are only 9 out of 131 participants that were from counties that are not the United States. Could you exclude these 9 participants from your analyses?
- It is unclear if the 131 participants responded to all the questions on your surgery. Were there any missing data? How did you deal with missing data if there is any? Could you provide one or two sentences to describe it. Could you included sample size in your regression tables?
- It’s unclear why regression analysis is needed for table 2, and which variables were independent variables and which were outcome variables. Is the pre COVID-19 health domain score the independent variable, and the difference between pre COVID-19 and post COVID-19 the outcome variable? In this case, I think the paired t-test is more appropriate to assess if there is significant difference before and after COVID-19 in these 6 health domains.
- In tables 3 and 4, it is unclear how you calculated the total variance explained by multiple variables. For example, table 3 shows that the following variables “Sleep and rest, Activities to satisfy sexual needs, Personal hygiene activities e.g., taking showers or baths, brushing teeth, grooming, and dressing, and Activities involving caring for children, other people, or pets” account for 19.96% of variance in total. Were these variables included in the same regression model? Was the 19.96% variance from the multivariable regression model. Or were these variables fitted in regression models separately, and 19.96% was the sum of the variance from different regression models? I think including these variables in the same model could provide a better prediction of the outcome variable, since there could be interactions between these variables.

Validity of the findings

no comment

·

Basic reporting

The manuscript adheres to the journal's guidelines by employing methods for data deidentification and upholding ethical standards. Furthermore, thorough scrutiny of the figures and tables has been undertaken. Overall, I commend the authors for their meticulous effort in presenting a meticulously structured research framework. The figures are pertinent and accurately labeled, facilitating a more lucid comprehension of the material. However, the manuscript still requires major revisions. .

Experimental design

The statistical analysis section lacks depth and explanation of the statistical models.
Line 219: It is unclear which variables were used as predictors for the Multiple linear regression model. Also, since the variables used in the study are mostly categorical, for example, Q14_5 in the data file has responses such as “Much better than before”, how did authors use these values in the context of regression model? Were they converted into a continuous scale? Were several MLR models built or just one? Using R2 values to “reflect the prediction in change” is not statistically appropriate. However, building a logistic regression model with two categories for the response variable (positive change and negative change) could have been more statistically sound. Any particular reason for high female participation in the study (86%)?
Overall, the premise for the study is promising but the statistical modeling employed is not fully comprehensible.

Validity of the findings

no comment

Additional comments

no comment

Reviewer 3 ·

Basic reporting

The authors propose a study investigating the effects of Covid-19 lockdowns on several aspect of occupational life, included overall quality of life, mental and physical health and social well-being. The manuscript is well written, and the literature is appropriate. I also found the overall structure of the manuscript organized in a clear scientific manner; tables and figures are also clear and simple.

Experimental design

The study aim is clearly stated and is in line with aims and scope of the Journal. The research question is well defined and meaningful. It also adds new data and interpretations about the effects of Covid-19 on individuals. The methods are clearly described and are sufficient for replicability.
My only main concern is about the reliability of answers provided by participants. Results would likely be different if the questionnaire was administered and answers provided during the lockdown, instead that at the end. Even if the spread of Covid 19 was still ongoing when authors collected data, it is also true that the first waves were very different from the later ones. Accordingly, the first waves had likely the strongest impact of mental, physical and overall quality of life of individuals. How the authors controlled for this aspect? Furthermore, a self-reported questionnaire as the limitation that not all the participants are supposed to have full awareness of themselves and of the world around them. this means that they may have exacerbated some aspects because of their psychological/psychiatric characteristics, rather than for "objective" reasons. The Covid-19 pandemic also unmasked several conditions of mental issue that may have influenced the results here reported.

Validity of the findings

The results are interesting and surely add information to a already important corpus of findings. Data sounds statistically appropriate; still it remains my concerns about the methodology and the reliability of answers provided by participants once the worst waves were finished.

---

## Round 0.2 · accepted · Accept

Given that you have addressed all the concerns raised by the reviewers, I believe the paper is ready for publication. Please follow the instructions regarding the formatting of the manuscript.

Reviewer 1 ·

Basic reporting

no comment

Experimental design

the authors have sufficiently addressed my comments and thus I have not further suggestions.

Validity of the findings

no comment

Reviewer 3 ·

Basic reporting

All my concerns have been addressed

Experimental design

All my concerns have been addressed

Validity of the findings

All my concerns have been addressed

Additional comments

All my concerns have been addressed